# Absence of Weak Localization Effects in Strontium Ferromolybdate

**Gunnar Suchaneck** [1,*] and **Evgenii Artiukh** [2]

1. TU Dresden, Institute of Solid-State Electronics, 10602 Dresden, Germany
2. SSPA (Scientific-Practical Materials Research Centre of NAS of Belarus), Cryogenic Research Division, 220072 Minsk, Belarus; sirfranzferdinand@yandex.ru
* Correspondence: gunnar.suchaneck@tu-dresden.de; Tel.: +49-351-46335281

**Abstract:** $Sr_2FeMoO_{6-\delta}$ (SFMO) double perovskite is a promising candidate for room-temperature spintronic applications, since it possesses a half-metallic character (with theoretically 100% spin polarization), a high Curie temperature of about 415 K and a low-field magnetoresistance (LFMR). The magnetic, resistive and catalytic properties of the double perovskite SFMO are excellent for spintronic (non-volatile memory), sensing, fuel cell and microwave absorber applications. However, due to different synthesis conditions of ceramics and thin films, different mechanisms of electrical conductivity and magnetoresistance prevail. In this work, we consider the occurrence of a weak localization effect in SFMO commonly obtained in disordered metallic or semiconducting systems at very low temperatures due to quantum interference of backscattered electrons. We calculate the quantum corrections to conductivity and the contribution of electron scattering to the resistivity of SFMO. We attribute the temperature dependence of SFMO ceramic resistivity in the absence of a magnetic field to the fluctuation-induced tunneling model. We also attribute the decreasing resistivity in the temperature range from 409 K to 590 K to adiabatic small polaron hopping and not to localization effects. Neither fluctuation-induced tunneling nor adiabatic small polaron hopping favors quantum interference. Additionally, we demonstrate that the resistivity upturn behavior of SFMO cannot be explained by weak localization. Here, the fitted model parameters have no physically meaningful values, i.e., the fitted weak localization coefficient ($B'$) was three orders of magnitude lower than the theoretical coefficient, while the fitted exponent ($n$) of the electron–electron interaction term ($C_n T^n$) could not be assigned to a specific electron-scattering mechanism. Consequently, to the best of our knowledge, there is still no convincing evidence for the presence of weak localization in SFMO.

**Keywords:** strontium ferromolybdate; electrical conductivity; weak localization

## 1. Introduction

Ordered double perovskite oxides of the general formula $A_2BB'O_6$ have drawn scientific interest for several decades to due to their interesting magnetic properties and their high potential for use in novel electronic devices. Recently, we published a review on resistivity and tunnel magnetoresistance in double perovskite [1]. Compounds with high spin polarization dramatically enhance spintronic device performance and are required for a new generation of non-volatile memories [2,3]. The room-temperature magnetization and large dielectric response of SFMO enable intriguing applications as microwave absorbers [4,5]. Superior catalytic properties, as well as stability in both oxidizing and reducing atmospheres, enable the use of SFMO not only as an anode but also as a cathode and in symmetrical solid oxide fuel cells. SOFC provides both electron and ionic conductivity, extending the triple-phase boundary compared to purely electronic or purely ionic conductors. These properties can be tailored to a particular application by the substitution of different metal cations at different sites on SFMO lattices. Moreover, iron-rich SFMO

materials are excellent catalysts in hydrocarbon oxidation and can prevent carbon deposition due to their ability to exchange lattice oxygen with the gaseous phase. They are also sulfur-tolerant. This opens the way to direct hydrocarbon-fueled SOFCs, eliminating the need for external fuel reforming and sulfur removal components. As such, SOFCs can be greatly simplified and operate with much higher overall efficiency, contributing to a solution to the lack of energy in our modern world [6]. The huge magnetic entropy change at the ferromagnetic transition makes SFMO a candidate for magnetocaloric cooling [7,8].

In a disordered electronic system, the electron motion is diffusive due to scattering at non-uniformities. Weak localization is a physical effect that occurs in disordered electronic systems at very low temperatures. This effect manifests itself as a correction ($\Delta\sigma$) to the conductivity (or, correspondingly, the resistivity) of a metal or semiconductor arising in cases in which the mean free path ($l$) is in the order of the wavelength ($\lambda_F = 2\pi/k_F$) of the carrier wavefunctions, i.e., $k_F l \sim 1$, with $k_F$ denoting the Fermi wave vector. The weak localization correction comes from quantum interference of backscattered electrons. Labeling all the trajectories according to the time ($t$) it takes for a classical particle to go around a loop, the classical probability ($dP$) that the diffusing particle returns into the phase volume ($dV$) at a given time ($t$) is given by [9] (p. 153)

$$dP = \frac{dV}{(4\pi Dt)^{d/2}} \tag{1}$$

where $P$ is the probability that a quantum mechanical particle will return to the starting point (the wave functions of both time-reversed paths constructively interfere with each other), $D = v_F^2\tau/d$ is the diffusion constant, $\tau$ is the elastic scattering time and $d$ is the dimension. The relevant phase volume can be estimated as $v_F\,dt\,(\delta\rho\delta\varphi)^{d-1}$ [10,11], where $v_F = (2E_F/m_e)^{1/2}$ is the Fermi velocity; $E_F$ is the Fermi energy; $m_e$ is the electron mass; $\delta\rho$ and $\delta\varphi$ characterize the transverse distance between electron trajectories 1 and 2 at the intersection point, respectively; and $\delta\varphi$ is the intersection angle between the two trajectories. For the interference between paths 1 and 2 to be effective, the uncertainty relation should hold:

$$p_F\delta\rho\delta\phi = \hbar k_F\delta\rho\delta\phi = h \tag{2}$$

where $p_F = m_e v_F$ is the Fermi momentum, and $h$ is the Planck constant. This leads to a relative change in conductivity of two- and three-dimensional systems, yielding [9] (p. 183):

$$\begin{aligned}\frac{\delta\sigma_{2D}}{\sigma_0} &= -\frac{2}{\pi k_F l_e}\ln\frac{l_\Phi}{l_e}, \\ \frac{\Delta\sigma_{3D}}{\sigma_0} &= -\frac{3}{2(k_F l_e)^2}\end{aligned} \tag{3}$$

where $\sigma_0$ is the residual low-temperature conductivity taken as the Drude conductivity, i.e., $\sigma_{\text{Dr}} = e^2 n_e \tau_e/m_e$, as determined by the relaxation time ($\tau$) of the dominating charge-scattering mechanism, $e$ is the electron charge, $n_e$ is the electron density, $\tau_e$ is the elastic scattering time and $m_e$ is the electron mass. In Equation (3), the negative sign is due to the fact that the returning trajectory should arrive at the intersection point with the momentum almost opposite to the initial trajectory. This means that interference lowers the conductivity. Note that for small sizes ($b$), the volume ($Dt)^{d/2}$ should be replaced by $(Dt)^{d/2}b^{3-d}$, since the charge carrier has a chance to diffuse repeatedly from one wall to the other, and the probability of finding it at any point across films or wires limited in size is the same [12]. However, we are considering macroscopic sizes rather than sizes of the order of atomic distances. The lower cutoff ($\tau_e$) is justified by the fact that within a given time ($\tau_e$), no elastic scattering occurs, resulting in no closed trajectories. The upper limit is given by [13]:

$$\tau_\Phi^{-1} \sim \left(\frac{T}{D^{d/2}N_0 a^{3-d}}\right)^{2/(4-d)}, \tag{4}$$

where $D = (v_F{}^2\tau_e/d)$ is the diffusion constant, $N_0$ is the one-spin density of states and $d$ is the characteristic sample dimension. The phase coherence time ($\tau_\phi$) defines the phase coherence length ($l_\phi = (D\tau_\phi)^{1/2}$). This yields conductivity corrections for two- and three-dimensional systems, which, in terms of quantum conductance ($e^2/h$), amounts to [9] (p. 282):

$$\Delta\sigma_{2D} = -\frac{2e^2}{\pi h}\ln\frac{l_\varphi}{l_e} = -\frac{e^2}{\pi h}\ln\left(\frac{l_\varphi}{l_e}\right)^2,$$
$$\Delta\sigma_{3D} = -\frac{e^2}{\pi h}\left(\frac{1}{l_e} - \frac{1}{l_\varphi}\right),$$

(5)

where $l_e$ is the mean free path of the electron.

Temperature modifies the phase coherence length ($l_\phi$), as well as the weak localization correction. The temperature dependence of $l_\varphi$ can be described as [14,15]

$$\frac{1}{l_\phi^2(T)} = \frac{1}{l_\phi^2(0)} + A_{ee}T^{p_1} + A_{ep}T^{p_2},$$

(6)

where $l_\varphi(0)$ is the zero-temperature phase coherence length, and $A_{ee}T^{p1}$ and $A_{ep}T^{p2}$ represent the contributions of two different dephasing mechanisms, e.g., electron–electron (ee) and electron–phonon (ep) interactions, respectively. Note that the temperature exponents $p_1$ and $p_2$ change fundamentally with the system dimensionality [16]. If electron–electron interaction is the dominant dephasing mechanism (also denoted as Nyquist dephasing mechanism), this gives rise to $\tau_{ee} \propto T^{-p}$, with values of $p$ equal to 0.66, and 1 for $d = 1$ and $d = 2$, respectively [9] (chapter 13.6.4), [13]. For $d = 3$, Equation (4) yields $p = 2$, while the result in [9] (p. 509) is $p = 1.5$. The electron–phonon interaction decoherence mechanism would provide a temperature dependence of the dephasing time ($\tau_\phi \propto T^{-3}$) [17]. Typically, one finds that $\tau_\varphi \approx \tau_{ep} \propto T^{-p}$ with the exponent of temperature $p \approx 2$–4 [16].

In metals and semiconductors at low temperatures, quasielastic (small energy transfer) electron–electron scattering is the dominant dephasing process. Assuming only one dephasing process and accounting for the fact that $l_\varphi(0)$ is in the order of 50 nm to several micrometers, that is, usually $l_\varphi(0) >> l_\varphi(T)$ [14,15,18–20], a simplified approximation formula is given by [19]:

$$\frac{1}{l_\phi^2(T)} \approx A'T^{p_1},$$

(7)

where $p_1 = 1$ represents quasielastic electron–electron scattering, in agreement with experimental results reported in $Bi_2Te_3$ thin films [21], 50 nm-thick $Cd_3As_2$ films [19], $Bi_2Te_3$ single crystals [15] and $Mo_xW_{1-x}Te_{2+\delta}$ ultrathin films [22]. As a result, for quasielastic electron–electron scattering, we obtain:

$$l_\varphi^2 = l_\varphi^2(T_0) \cdot \frac{T_0}{T},$$

(8)

Consequently, Equation (5) transforms to

$$\Delta\sigma_{2D,WL} = -\frac{e^2}{\pi h}\ln\left(\frac{l_\phi^2(T_0)}{l_e^2} \cdot \frac{T_0}{T}\right) = \frac{e^2}{\pi h}\ln\left(\frac{l_e^2}{l_\phi^2(T_0)} \cdot \frac{T}{T_0}\right)$$
$$= \frac{e^2}{\pi h}\left[\ln\left(\frac{l_e^2}{l_\phi^2(T_0)}\right) + \ln\left(\frac{T}{T_0}\right)\right]$$
$$\delta\sigma_{3D,WL} = -\frac{e^2}{\pi}\left(\frac{1}{l_e} - \frac{1}{l_\phi}\right) = -\frac{e^2}{\pi \cdot l_e} + \frac{e^2}{\pi \cdot l_\phi(T_0)}\left(\frac{T}{T_0}\right)^{1/2}$$

(9)

Finally, the total bulk conductivity consisting also two-dimensional contributions under high magnetic fields is written as:

$$\sigma(T) = \sigma_0 + A \cdot \ln\left(\frac{T}{T_0}\right) + B \cdot \left(\frac{T}{T_0}\right)^{1/2},$$

(10)

where

$$\sigma_0 = \sigma_D - \frac{e^2}{\pi h b} \cdot \ln\left(\frac{l_\phi^2(T_0)}{l_e^2}\right) - \frac{e^2}{\pi h \cdot l_e}, \tag{11}$$

and the coefficients:

$$A = \frac{e^2}{\pi h b}, \; B = \frac{e^2}{\pi h \cdot l_\phi(T_0)}, \tag{12}$$

where $b$ is the effective thickness of the 2D layer introduced to convert a 2D conductivity to a 3D conductivity. A minimum of conductivity appears only when the second right-side term in Equation (10) changes sign, i.e.,

$$B = -\frac{2A}{(T_{max}/T_0)^{1/2}} \underset{T_0 = T_{max}}{=} -2A, \tag{13}$$

with regard to Equation (10), quantum correction to residual conductivity may be written as [23]:

$$\Delta\sigma(T) = A\prime \cdot \ln T + B\prime \cdot T^{1/2}, \tag{14}$$

Assuming that Matthiessen's rule holds and representing resistivity as a sum of elastic and inelastic contributions, where the latter increase with increasing temperature due to a power law (e.g., a term such as $CT^n$), the resistivity data can be fitted to [24,25]:

$$\rho(T) = \frac{1}{\sigma_0 + A\prime \cdot \ln T + B\prime \cdot T^{1/2}} + C_n T^n, \tag{15}$$

where $n$ is equal to $3/2$, 2, 3, depending on the dominant scattering mechanism [26].

Separate conductivity quantum correction terms in the form of $A'\ln T$ and $B'T^{1/2}$ were introduced by Kumar et al. for $La_{0.7}Ca_{0.3}MnO_3$ [27]. The first term ($A'\ln T$) was also attributed also to the Kondo effect [28]. In order to account for higher-order scattering mechanisms and to extend the analytical description to higher temperatures, an additional term ($CT^n$) was included. The electron–electron interaction term ($B'T^{1/2}$) was also applied to $La_{0.7-x}Y_xSr_{0.3}MnO_3$ ($0 \le x \le 0.2$) ceramics with a thickness of 1 mm [29], $La_{0.7}A_{0.3}MnO_3$ (A = Ca, Sr, Ba) ceramics [30] and $La_{0.6}Re_{0.1}Ca_{0.3}MnO_3$ (Re = Pr, Sm, Gd, Dy) ceramics with a thickness of 1 mm—all three of which were prepared by conventional solid-state reaction [31]—as well as to $SrRuO_3$ thin films with a thickness of 6 to 8 nm deposited by pulsed laser deposition onto (100)$SrTiO_3$ substrates [24], ultrathin $La_{0.7}Sr_{0.3}MnO_3$ films (3.5 to 40 nm) deposited by molecular beam epitaxy [25] and metallic $SrRuO_3$ and $LaNiO_3$ thin films deposited by pulsed laser deposition with a thickness of 6 and 240 nm, respectively [32].

In the case of $La_{0.7}Sr_{0.3}MnO_3$ ultrathin film, crossover from $T^{1/2}$ to $\ln T$ behavior in the low-temperature resistivity dependence with decreasing thicknesses was attributed to a change in the dimensionality of the system, from 3D for samples thicker than 20 nm to 2D in the limit of ultrathin samples. The origin of this effect is that in the case of a film thickness greater than the Landau orbit length ($L_H = (\hbar/2eB)^{1/2}$), the system behaves essentially as 3D, while the opposite electron confinement results in 2D behavior of the system. In the case of 3D metallic and ferromagnetic $SrRuO_3$ and metallic and paramagnetic $LaNiO_3$ epitaxial thin films, the term $B'T^{1/2}$ was split into two terms; in the 3D case, the first term ($b_1T^{p/2}$) accounts for the weak localization, and the $b_2T^{1/2}$ term stands for the renormalized electron–electron interaction quantum corrections [33]. In the 2D case, both these quantum corrections to conductivity have a similar temperature dependence. Here, only an $\ln T$ term remains. The quantum correction of conductivity in metallic $LaNi_{1-x}Co_xO_3$ ($0 \le x \le 0.75$) below 2 K follows a power law ($BT^m$), where, away from the metal–insulator transition ($x \le 0.4$), $m$ takes a value of $m = 0.3 \ldots 0.4$. Such power-law conductivities are also observed on the metallic side of the metal–insulator transition for other $ABO_3$ oxides [34].

In this work, we attributed the temperature dependence of the SFMO ceramic resistivity in the absence of a magnetic field to the fluctuation-induced tunneling model and the

decrease in resistivity in the temperature range of 409 K to 590 K and not to localization effects but to adiabatic small polaron hopping. Neither fluctuation-induced tunneling nor adiabatic small polaron hopping favors quantum interference. We also demonstrate that the resistivity upturn behavior of SFMO cannot be explained by the weak localization effect.

## 2. Methods

First, we estimate the coefficient ($A'$) in Equation (15). Taking $l_\Phi < 0.46$ nm at room temperature [35] and $b = 20$ nm [25], the ratio of the coefficients $A'/B'$ are in the order of $10^{-2}$. Because for arguments larger than one, the natural logarithm function is smaller than the root function, the logarithmic term can be neglected. A further indication of small $A'$ coefficients arises from the depth of the conductance minimum, i.e., the difference between $\rho(0)$ and $\rho(T_{min})$, which changes with increasing magnetic flux [27]. Values larger than 35 m$\Omega$·cm in the absence of a magnetic field and down to 2.4 m$\Omega$·cm at 7 T [36] yield coefficients of $A' = 10^{-2} \cdot \rho(0) - \rho(T_{min}))$ [28,37]. In La$_{0.7}$Sr$_{0.3}$MnO$_3$ ultrathin films (3.5 to 40 nm) deposited by molecular beam epitaxy [25], the agreement of fits of conductivity data to Equation (15) with only the $A'\ln T$ term ($B' = 0$) was much worse than in the case with only the $B'T^{1/2}$ term ($A' = 0$). Therefore, in the following, we neglect the term $A'\ln T$. This corresponds to the common practice of describing similar materials, such as La$_{0.7-x}$Y$_x$Sr$_{0.3}$MnO$_3$ ($0 \le x \le 0.2$) ceramics with 1 mm thickness prepared by conventional solid-state reaction [29], La$_{0.7}$A$_{0.3}$MnO$_3$ (A = Ca, Sr, Ba) ceramics prepared by conventional solid-state reaction [30] and SrRuO$_3$ thin films with a thickness of 6 to 8 nm deposited by pulsed laser deposition onto (100)SrTiO$_3$ substrates [24].

The coefficient $B'$ is given by [23]:

$$B\prime = 0.0309 \cdot \frac{e^2}{\hbar} \cdot \sqrt{\frac{k}{\hbar D}} = \frac{2.72149 \cdot \Omega^{-1}\mathrm{K}^{-1/2}\mathrm{s}^{-1/2}}{\sqrt{D[\mathrm{m}^2/\mathrm{s}]}}. \tag{16}$$

where $\hbar = h/2\pi$ is the Planck constant expressed in J s radian$^{-1}$. For a mean free path ($l_e$) of 0.46 nm at room temperature [35] and a carrier relaxation time ($\tau$) of $1.6 \times 10^{-14}$ s [38], the carrier diffusion constant is $1.32 \times 10^{-5}$ m$^2$ s$^{-1}$, yielding a co-efficient of $B' \approx 748$ $\Omega^{-1}$ m$^{-1}$ K$^{-1/2}$, in satisfactory agreement with values of $B' \approx 360 \dots 500$ $\Omega^{-1}$ m$^{-1}$ K$^{-1/2}$ in (Ni$_{0.5}$Zr$_{0.5}$)$_{1-x}$Al$_x$ metallic glasses [39] and with a universal value of $B' \approx 600$ $\Omega^{-1}$ m$^{-1}$ K$^{-1/2}$ of amorphous and disordered metals [40].

The coefficient $C_n$ is calculated as follows. We start with Drude conductivity:

$$\rho_D = \frac{\hbar k_F}{n_e e^2 l_e}, \tag{17}$$

where $n_e$ is the electron density, and $e$ is the electron charge. Taking $n_e = 1.1 \times 10^{28}$ m$^{-3}$ [41], we obtain $\hbar k_F = 4.8 \times 10^{-25}$ Jsm$^{-1}$ and

$$\rho_D = \frac{1.87 \cdot 10^{-15} \Omega\mathrm{m}}{l_e[\mathrm{m}]}, \tag{18}$$

We assume electron mean free paths of $l_e = 0.46$ nm and 1.11 nm at room temperature and 4 K, respectively, calculated according to the ordinary Hall coefficient [35]. This yields $\rho_D = 4.07$ $\mu\Omega$m at room temperature and $\rho_D = 1.68$ $\mu\Omega$m at 4 K, in satisfactory agreement with experimental data on single-crystal SFMO reported in [42]. Now, we assume that in a ferromagnetic state below room temperature, magnetic scattering controls electrical transport in SFMO at low temperatures [35]. The mean free path of electrons scattered

by a spin wave with energy ($E_\sigma$) traveling through the bcc I4/mmm lattice in thermal equilibrium at a temperature ($T$) is given by [43]:

$$l_e = \frac{S^2 V^{1/3} \theta^{-7/2}}{\pi \zeta(3/2)(E_\sigma/kT)} \approx 0.38 \cdot S^{9/2} a \cdot \left(\frac{J}{kT}\right)^{5/2},$$ (19)

where $S$ is the effective spin ($S_{eff} = (S_{Fe} S_{Mo})^{1/2} = (1/2 \cdot 5/2)^{1/2} = 1.118$), $V$ is the unit cell volume, $\theta$ is a dimensionless temperature, $\zeta$ is the Rieman zeta function, $a$ is the lattice constant and $J$ is the exchange constant of the 180° Fe–O–Fe interaction, amounting to $-25$ K [44,45]. Finally, we arrive at:

$$\rho_D = 2.413 \cdot 10^{-9} \cdot T^{5/2} \Omega m,$$ (20)

Compared to the experimental value of $R_{2.5} = 1.4 \times 10^{-11}$ $\Omega m K^{-5/2}$ in a relation ($\rho = \rho_0 + R_{2.5} T^{2.5}$), the calculated value of $R_{2.5}$ is overestimated by almost two orders of magnitude, in part due to the approximation of the Fermi surface as a sphere and the disregard of additional $s$–$d$ transitions in transition metals, which reduce the mean free path [1]. The value of $R_{2.5}$ may also be lowered by assuming a higher effective spin.

## 3. Results and Discussion

Strontium ferromolybdate ($Sr_2FeMo_{6-\delta}$, SFMO) is a half-metallic, ferrimagnetic compound with a saturation magnetization of 4 $\mu_B$/f.u [36]. However, SFMO does not exhibit a general metallic conductivity mechanism. In the absence of a magnetic field, the temperature dependence of the conductivity of SFMO ceramics [36] is well-described by the fluctuation-induced tunneling (FIT) model 46, e.g., by the presence of conducting grains separated by nanosized energy barriers where large thermal voltage fluctuations occur when the capacitance of an intergrain junction is in the order of 0.1 fF. Here, tunneling occurs between large metallic grains via the intergrain junctions with a width ($w$) and area ($A$). The FIT model is specified by three parameters [46]: (i) the temperature ($T_1$) characterizing the electrostatic energy of a parabolic potential barrier,

$$kT_1 = \frac{A \cdot w \cdot \varepsilon_o E_0^2}{2},$$ (21)

where $k$ is the Boltzmann constant and the characteristic field ($E_0$) is determined by:

$$E_0 = \frac{4V_0}{e \cdot w};$$ (22)

(ii) the temperature ($T_0$) representing $T_1$ divided by the tunneling constant,

$$T_0 = T_1 \cdot \left(\frac{\pi \chi w}{2}\right)^{-1},$$ (23)

with the reciprocal localization length of the wave function

$$\chi = \sqrt{\frac{2m^* V_0}{\hbar^2}},$$ (24)

where $m^*_e$ is the effective electron mass; and (iii) the residual resistivity ($\rho_0$). The resulting resistivity of this model is then given by [46]:

$$\rho(T) = \rho_0 \exp\left(\frac{T_1}{T_0 + T}\right),$$ (25)

In our case, the model parameters amount to $T_0 = 141.1$ K, $T_1 = 25.6$ K and $\sigma_0 = 1/\rho_0 = 37.16$ S/cm (cf. Figure 1). The FIT model was recently applied to intergrain tunneling in polycrystalline $Sr_2CrMoO_6$ and $Sr_2FeMoO_6$ ceramics [47], half-metallic

double perovskite $Sr_2BB'O_6$ (BB′–FeMo, FeRe, CrMo, CrW, CrRe) ceramics [48] and in $Ba_2FeMoO_6$ thin films [49].

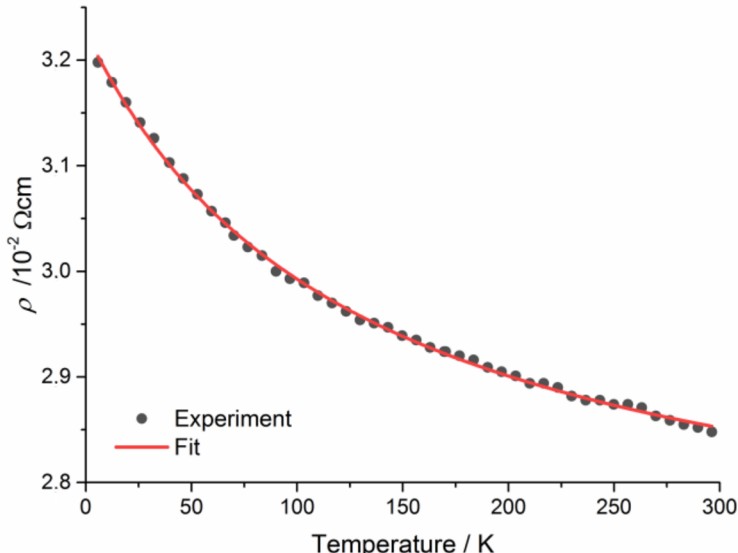

**Figure 1.** Fitting of the conductivity data of SFMO ceramics at zero magnetic flux [36] to the fluctuation-induced tunneling model [46]. Fitting parameters are $T_0$ = 141.1 K, $T_1$ = 25.6 K and $\sigma_0$ = 37.16 S/cm.

One feature attributed to weak localization in SFMO ceramics is the decrease in resistivity in the temperature range between 405 K and 590 K [50]. This resistivity behavior is considered more in detail below. According to [50], the resistivity behavior of vacuum-annealed SFMO ceramics above room temperature is separated into three regions: (i) from 300 K up to the Curie temperature of about 405 K, the electrical resistivity increases with temperature and shows metallic behavior; (ii) above 405 K up to approximately 590 K, the resistivity decreases with temperature; and (iii) finally, from 590 K up to 900 K, the resistivity increases as the material becomes metallic again [50].

Another report on the electrical resistivity of SFMO indicates metallic behavior up to 420 K, a decrease in resistivity in the temperature range of 420–820 K and reversion to metallic behavior between 820 and 1120 K [51]. A similar resistivity behavior with a resistivity maximum at about 450 K—far above the Curie temperature of ~330 K [52]—was obtained for $Ba_2FeMoO_{6-\delta}$, while $Ca_2FeMoO_{6-d}$ shows solely metallic behavior in the whole temperature range of 320–1120 K [51].

A more detailed consideration of the reported resistivity behavior of $Sr_2FeMoO_{6-\delta}$ and $Ba_2FeMoO_{6-\delta}$ above $T_C$ [50,51] reveals a convincing fit to the adiabatic small polaron hopping model [53] (Figure 2). In the small polaron model, electrical conduction of perovskites at higher temperatures, i.e., above a certain transition temperature, occurs as a result of small polarons moving through the lattice by thermally activated jumps between neighboring sites. The transition temperature from small polaron motion in a conduction band to small polaron hopping was estimated to be in the order of $\theta_D/2$ (with $\theta_D$ the Debye temperature), which amounts to 338 K for SFMO [54]. The adiabatic small polaron hopping model yields a resistivity of [53]

$$\rho = \rho_0 T \exp\left(\frac{E_a}{kT}\right), \tag{26}$$

where $E_a$ is the thermal activation energy. The obtained $E_a$ values are 0.045–0.08 eV for $Sr_2FeMoO_{6-\delta}$ and about 0.13 eV for $Ba_2FeMoO_{6-\delta}$ and are in the order of the values of other perovskites and double perovskites, i.e., $La_{1-x}Sr_xCo_{1-y}Fe_yO_3$ [55], $Sr_{1.6}Sm_{0.4}MgMoO_{6-\delta}$,

$Sr_{1.4}Sm_{0.6}MgMoO_{6-\delta}$ and $Sr_{1.2}Sm_{0.8}MgMoO_{6-\delta}$ [56], as well as $Sr_2Fe_{1.5}Mo_{0.5}O_{6-\delta}$ and $Sr_2Fe_{1.5}Mo_{0.5-x}Nb_xO_{6-\delta}$ [57].

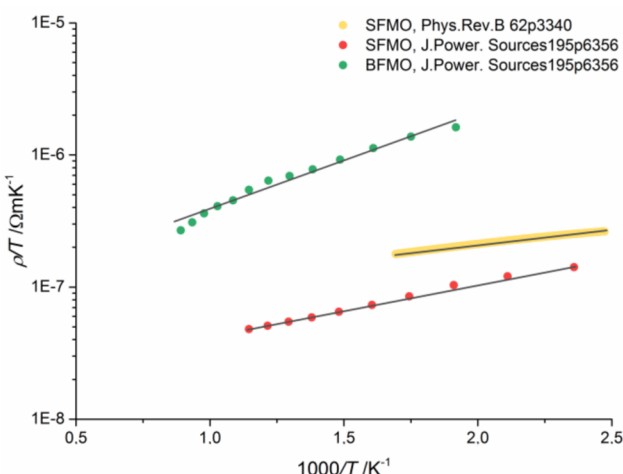

**Figure 2.** Fitting of the conductivity data of $Sr_2FeMoO_6$ ceramics between 405 K and 590 K [50], as well as those of $Sr_2FeMoO_6$ ceramics between 420 and 870 K and $Ba_2FeMoO_6$ ceramics between 520 and 1120 K [51], to the adiabatic small polaron hopping model [53].

Weak localization effects were taken into account in order to explain the presence of minima in the $\rho$–$T$ curves of perovskite oxides exhibiting metallic conductivity [25,27,33,58,59]. It has been suggested [29,30] that the resistivity minimum and, consequently, the resistivity upturn at lower temperatures arise from the competition of two contributions—one usual increase combined with a decrease with the increase in the temperature of the other. In Equation (15), the corresponding terms are $(B'\ln T)^{-1}$ and $C_nT^n$. The resistivity-versus-temperature plots of SFMO ceramics in a magnetic field [36] are very similar to those of $La_{0.7}Ca_{0.3}MnO_3$ and $La_{0.7}Sr_{0.3}MnO_3$ thin films [25,27]. A resistivity minimum also appears in polycrystalline SFMO ceramics at low temperatures [36], which was explained by weak localization [59]. In this case, a quantum correction term $(A_WT^{p/2})$ [25,59] was added to the residual resistivity $(\sigma_0)$, and an electron interaction term $(A_pT^n)$ was added to the resistivity. A similar approach was applied to perovskite ceramics ($LaNi_xCo_{1-x}O_3$ and $Na_xTa_yW_{1-y}O_3$ [60], $La_{0.5}Pb_{0.5}MnO_3$ and $La_{0.5}Pb_{0.5}MnO_3$ ceramics containing 10 at.% Ag in a dispersed form [23]).

To evaluate the origin of the low-temperature resistivity minimum in SFMO, we fitted the experimental data [36] to Equation (15), assuming $A' = 0$ (Figure 3, Table 1). The increase in the $\sigma_0(B)$ values corresponds to a negative magnetoresistance obtained in ferrimagnetic SFMO ceramics, which arises due to the suppression of spin disorder by the magnetic field [35,36,50]. The fitted $\sigma_0$ values describe a power-law magnetic flux dependence of the magnetoresistance ($-MR \propto B^m$) with a power of $m = 0.284$. This value lies in between the values of $m = 0.5$ for metallic behavior and electron–electron interaction in the temperature range between the temperature of minimum resistivity and the Curie temperature and $m = 0.1$ for semiconducting conductivity behavior, both at high magnetic fluxes [61]. The fitted $B'$ values are three orders of magnitude lower than those calculated above. Consequently, the elastic scattering time should be reduced by six orders of magnitude. This is physically nonsensical. Furthermore, the $C_nT^n$ term does not correspond to physically meaningful quantities. The values of the exponent ($n$) cannot be attributed to a specific electron-scattering mechanism. On the other hand, electron scattering should sufficiently change depending on the magnetic flux. When assuming $n = 2$ for electron–electron scattering and assuming $C_n = 2.16 \times 10^{-11}$ $\Omega mK^{-2}$ from [41], the fit becomes of much worse quality (Figure 4). Here, a satisfactory fit occurs only for $B = 0$. Assuming $n = 2.5$ and the value of $C_n$ in Equation (20), the fit is even worse, with no satisfactory fit, even for $B = 0$.

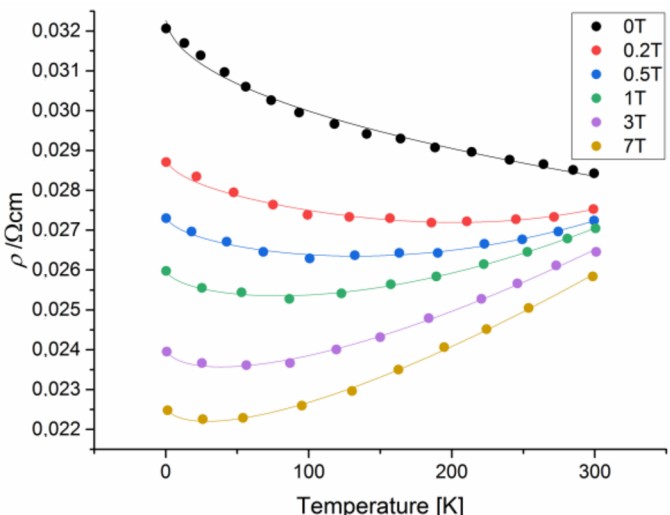

**Figure 3.** Fit of the conductivity data of $Sr_2FeMoO_6$ [36] to Equation (15) assuming $A' = 0$.

**Table 1.** Fit of the parameters of Equation (15) to the experimental data reported in [36] assuming $A' = 0$.

| $B$, T | $\sigma_0$, Sm$^{-1}$ | $B'$, Sm$^{-1}$K$^{-1/2}$ | $C_n$, $\Omega$mK$^{-n}$ | $n$ |
|---|---|---|---|---|
| 0 | 3087 | 0.222 | $-1.37 \times 10^{-5}$ | 0.61 |
| 0.2 | 3464 | 0.174 | $6.09 \times 10^{-12}$ | 3.31 |
| 0.5 | 3649 | 0.161 | $3.75 \times 10^{-9}$ | 2.29 |
| 1 | 3839 | 0.166 | $1.15 \times 10^{-7}$ | 1.77 |
| 3 | 4150 | 0.235 | $2.21 \times 10^{-6}$ | 1.34 |
| 7 | 4414 | 0.270 | $3.73 \times 10^{-6}$ | 1.22 |

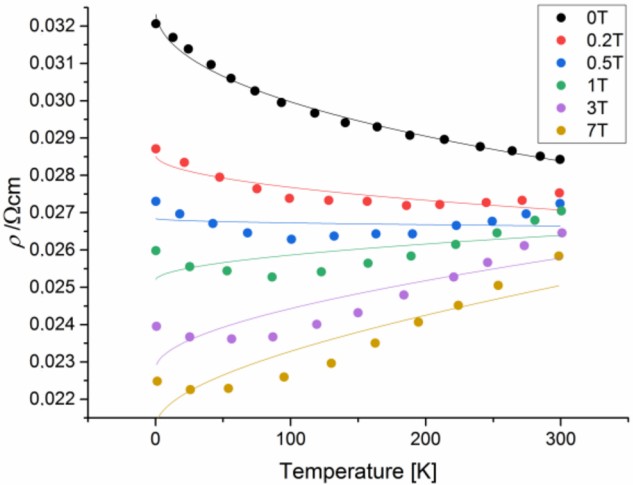

**Figure 4.** Fit of the conductivity data of $Sr_2FeMoO_6$ [36] to Equation (15) assuming $A' = 0$, $C_n = 2.16 \times 10^{-11}$ and $n = 2$.

Thus, the only possible conclusion is that the resistivity upturn at low temperatures in $Sr_2FeMoO_6$ ceramics cannot be modeled by weak localization correction due to quantum interference. Note that neither fluctuation-induced tunneling nor adiabatic small polaron hopping provides favorable conditions for quantum interference of backscattered electrons.

## 4. Conclusions

Herein, we related the temperature dependence of SFMO resistivity in the absence of a magnetic field to the fluctuation-induced tunneling model. The decrease in resistivity

above the resistivity maximum around the Curie temperature is attribute to adiabatic small polaron hopping instead of localization effects. Neither fluctuation-induced tunneling nor adiabatic small polaron hopping favors quantum interference. This is evidenced by the observation that the resistivity upturn behavior of SFMO cannot be explained by weak localization. Here, the fitted model parameters have no physically meaningful values, i.e., the fitted weak localization coefficient ($B'$) was three orders of magnitude lower than the theoretical value, while the fitted exponents ($n$) of the electron–electron interaction term ($C_n T^n$) could not be assigned to a specific electron-scattering mechanism. Consequently, to the best of our knowledge, there is still no convincing evidence for the presence of weak localization in SFMO.

**Author Contributions:** Conceptualization, G.S.; methodology, G.S.; software, E.A.; validation, G.S. and E.A.; formal analysis, G.S.; investigation, G.S. and E.A.; resources, G.S.; data curation, E.A.; writing—original draft preparation, G.S.; writing—review and editing, E.A.; visualization, E.A.; supervision, G.S.; project administration, G.S.; funding acquisition, G.S. All authors have read and agreed to the published version of the manuscript.

**Funding:** This work was funded by the EU project H2020-MSCA-RISE-2017-778308-SPINMULTIFILM.

**Institutional Review Board Statement:** Not applicable.

**Informed Consent Statement:** Not applicable.

**Data Availability Statement:** The original contributions presented in this study are included in the article. Further inquiries can be directed to the corresponding author.

**Acknowledgments:** The authors thank N. Sobolev (University Aveiro) for valuable discussions on the topic of this work.

**Conflicts of Interest:** The authors declare no conflict of interest.

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
