# Peer review of "Absence of Weak Localization Effects in Strontium Ferromolybdate"

_applsci, doi:10.3390/app13127096_

Round 1

Reviewer 1 Report

In the review of research article, titled: Absence of weak localization effects in strontium ferromolybdate by Suchaneck. Overall the presentation of the research work is good but there exist few issues, which are needed to be addressed before being published.

1.       In the abstract, authors should add a sentence clearly presenting the application where this study can be implemented.

2.       Introduction portion is well presented, but citation is not made from the latest published articles for this work.

3.       What is the reason of decrement in the resistivity with effect to varying temperature in Figure 1.

4.       English should be revised with the help of the native English speaker.

English quality is much better, little improvement is required.

Author Response

In the review of research article, titled: Absence of weak localization effects in strontium ferromolybdate by Suchaneck. Overall the presentation of the research work is good but there exist few issues, which are needed to be addressed before being published.

1. In the abstract, authors should add a sentence clearly presenting the application where this study can be implemented.

We added: The magnetic, resistive and catalytic properties of the double perovskite SFMO are excellent for spintronic, sensing, fuel cell and microwave absorbing applications. A more detailed description of SFMO applications is now given in the introduction.

2. Introduction portion is well presented, but citation is not made from the latest published articles for this work.

The introduction was focused to the effect of weak localization. Here, publications concerning SFMO are scarce (2 publications from the same group). Now, a break describing SFMO application was added including the most important recent articles in this field.

3. What is the reason of decrement in the resistivity with effect to varying temperature in Figure 1.

According to the fluctuation induced tunneling (FIT) model, with increasing temperature, the denominator T + T0 increases. Thus, the argument of the exponent becomes smaller with increasing temperature lowering the resistivity. The physical origin is the decrease of the intergrain tunneling current with temperature.

4. English should be revised with the help of the native English speaker.

English grammar is now revised by a native speaker.

Reviewer 2 Report

check the file

there is some typos, plz check it.

Author Response

1. In the abstract, it is mentioned that "Sr2FeMoO6-δ (SFMO) double perovskite is a promising candidate for room-temperature spintronic applications since it possesses a half-metallic character (with theoretically 100% spin polarization), a high Curie temperature of about 415 K, and a low-field magnetoresistance (LFMR).” It can be seen that Sr2FeMoO6-δ (SFMO) double perovskite is a promising candidate for room performance spintronic applications, so why did you choose Sr2FeMoO6-δ (SFMO) double perovskite in this article for research at high or low temperatures ?

The promising for device application properties of SFMO are temperature dependent. For a given application, the properties in a certain temperature range are of interest. For spintronic application the benefit of SFMO is the still high enough spin polarization and room temperature and the comparable high Curie temperature. For magnetic field sensing the corresponding benefit is the large low-field magnetoresistance at liquid nitrogen temperatures, for solid oxide fuel cells the stability and the high catalytic activity in the temperature range 600-800°C. This paper considers weak localization effects which occur independent on temperature but depend on the electron mean free path which is strongly temperature dependent. On the other hand, evidence of weak localization in manganites is given by the presence of minima in the ρ–T curves. Thus we have investigated the temperature range where ρ vs T data is available.

2. Your title is "Absence of weak localization effects in strontium ferromolybdate", but in the abstract it is mentioned that "In this work, we consider the weak localization effect in SFMO occurrence in disordered metallic or semiconducting systems at very low temperatures due to quantum interference of back-spotted electrons. This seems to be contradictory.

This piece of text was misleading which is now corrected in the text: In this work, we consider the occurrence of a weak localization effect in SFMO commonly obtained in disordered metallic or semiconducting systems at very low temperatures due to quantum interference of back-scattered electrons.

3. I suggest you read some new works, such as: 10.1002/pssb.202200012 and 10.1007/s11467-022-1199-5 (MA2Z4).

The paper 10.1002/pssb.202200012 is our recent review paper, now included in the introduction as Ref. 1. The paper 10.1007/s11467-022-1199-5 is devoted to the catalytic properties of MA2Z4 (M = Sc-Zn, Y-Ag, Hf-Au; A=Si, Ge; Z=N, P) monolayers which should be discussed for applications as photoelectric or photocatalytic materials what is not in the focus of this paper.

4. In Figure 2, there is a line without dot. Plz check it.

The data in the work of D. Niebieskikwiat et al. are so closely that they form a quasi-line. Taking empty circles as symbol, the figure does not become less confusing.

5. In abstract and conclusion part, we need some exact data or result. Especially, in the conclusion, there is no exact number or highlight.

We added text: Here, the fitted model parameters have no physically meaningful values, i.e. the fitted weak localization coefficient B´ was three orders of magnitude lower than the theoretical one while the fitted exponents n of the electron-electron interaction term CnTn could not be assigned to a specific electron scattering mechanism.

6. Formula 14, 15 and 16 have formatting errors and need to be edited again.

We have made small corrections in the formulas. Probably, the obtained formatting errors are due to problems displaying MathType equations. We will pay attention to this problem again during the proof correction.